# Pathogenic Single Nucleotide Polymorphisms on Autophagy-Related Genes

**DOI:** 10.3390/ijms21218196

**Published:** 2020-11-02

**Authors:** Isaac Tamargo-Gómez, Álvaro F. Fernández, Guillermo Mariño

**Affiliations:** 1Instituto de Investigación Sanitaria del Principado de Asturias, 33011 Oviedo, Spain; tamargoisaac@uniovi.es; 2Departamento de Biología Funcional, Universidad de Oviedo, 33011 Oviedo, Spain

**Keywords:** autophagy, pathology, SNPs, variants, polymorphisms, ATGs, autophagic receptors, lysosomes

## Abstract

In recent years, the study of single nucleotide polymorphisms (SNPs) has gained increasing importance in biomedical research, as they can either be at the molecular origin of a determined disorder or directly affect the efficiency of a given treatment. In this regard, sequence variations in genes involved in pro-survival cellular pathways are commonly associated with pathologies, as the alteration of these routes compromises cellular homeostasis. This is the case of autophagy, an evolutionarily conserved pathway that counteracts extracellular and intracellular stressors by mediating the turnover of cytosolic components through lysosomal degradation. Accordingly, autophagy dysregulation has been extensively described in a wide range of human pathologies, including cancer, neurodegeneration, or inflammatory alterations. Thus, it is not surprising that pathogenic gene variants in genes encoding crucial effectors of the autophagosome/lysosome axis are increasingly being identified. In this review, we present a comprehensive list of clinically relevant SNPs in autophagy-related genes, highlighting the scope and relevance of autophagy alterations in human disease.

## 1. Introduction

Sequence variations are the basis of genetic heterogeneity, which is essential for species to improve their fitness in the environment. Among these genetic changes, alterations with a minor allele frequency of at least 1% in a given population are called polymorphisms, and variants that affect only one base of the sequence (including exchanges, deletions or insertions) are termed single nucleotide polymorphisms (SNPs). From a clinical perspective, these variants are important because they alter the activity of the affected gene products. In other words, SNPs can be the underlying origin of different types of diseases or even explain the differential effect of some treatments in determined individuals. SNPs are mainly identified and analyzed by genome-wide association studies (GWASs), unveiling alleles that determine the susceptibility of their carriers to a given condition. It is not surprising that variants that negatively impact the function of genes involved in essential processes, such as DNA repair or autophagy, have been widely associated with different pathologies.

Tightly integrated into the cellular network of the stress response, autophagy is an essential mechanism for the maintenance of cellular homeostasis [1]. This catabolic pathway, present in all nucleated cells, can be defined as any mechanism which mediates the degradation of cellular components, including entire organelles, by the action of lysosomal hydrolases (in fact, “autophagy” derives from Greek, meaning “self-eating”) [2]. According to the way the cargo is transferred to the lysosome, there are three main autophagic pathways, namely microautophagy, chaperone-mediated autophagy (CMA) and macroautophagy [3]. During microautophagy, for example, substrates are directly engulfed by lysosomal protrusions or invaginations [4], while CMA requires the activity of chaperone Hsc70 (HSPA8) and LAMP2A to selectively recognize and internalize proteins showing the KFERQ motif [5]. In contrast, macroautophagy (which is the focus of this review, and will be hereafter referred to as “autophagy”) is based on the sequestration of cytoplasmic content by double-membrane vesicles, termed “autophagosomes” [6]. Once fully formed, autophagosomes eventually fuse their outer membrane with membranes of acidic lysosomes to become autolysosomes. Autolysosomes have hydrolytic activity, degrade their cargo, and recycle essential biomolecules to the cytoplasm [7]. Autophagy is active in all eukaryotic cells at basal rates, allowing the periodic renovation of cytosolic components or cytoplasmic organelles, acting as a housekeeping mechanism to preserve homeostasis. However, in response to a variety of cellular stresses, including nutrient deprivation, hypoxia, the accumulation of damaged organelles, protein aggregates or the presence of intracellular pathogens, the rate of autophagic degradation increases. This allows cells to eliminate damaged or harmful components through catabolism while supplying nutrients and energy to preserve cell viability. Given its fundamental roles in cell physiology, it was hypothesized early on that autophagy dysregulation could contribute to the pathogenesis of different diseases. Accordingly, a growing number of studies have linked autophagy alterations to a wide range of human pathologies, from immunological disorders to neurodegeneration or cancer [8] (Figure 1).

In this review, we collect and go through a comprehensive list of SNPs in autophagy-related genes that have been associated with human diseases. This approach highlights the relevance of specific genes and variants in human pathology, while giving us new insights into the real scope of autophagy-related SNPs in disease. We include variants in genes whose products are part of the main autophagy core machinery and also those in genes codifying lysosomal proteins, as its disruption leads to the alteration of the autophagosome/lysosome axis.

## 2. Autophagy Dysregulation in Disease

As it has been previously reviewed, several human disorders show alterations in autophagy [8,9]. Pathological dysregulation of autophagy is not restricted to a specific type of disease. In fact, a wide range of disorders that may not share a common etiology and affect different tissues or organs have been connected to autophagy dysregulation [10] (Figure 1). However, how autophagy alteration contributes to the pathogenesis of specific pathologies is far from being completely understood, as important questions remain unanswered [11].

One of the clinical fields in which autophagy’s role has been more extensively studied is oncology. Although it is still a long-standing subject of debate in the field, it is becoming increasingly clear that autophagy plays a dual role in this context, either favoring or fighting against the development of cancer [12]. On the one hand, the protective activity of this pathway prevents the initial malignant transformation of cancer cells. On the other hand, once the tumor is formed, autophagy would help the cancerous cells survive. All in all, the beneficial or detrimental effect of autophagy in cancer is likely to be tumor- and stage-dependent. Unsurprisingly, pathogenic and protective SNPs have been reported in autophagy-related genes, and a growing number of studies have been profiling the autophagy-related gene prognostic signature of different types of cancer [13,14,15,16,17,18,19,20,21,22,23,24]. The pivotal role of antitumor immunity against cancer progression adds even more complexity to the autophagy–cancer relation, as autophagy is also important for immunological processes. 

In fact, autophagy plays an essential role in the correct functioning of the immune system, acting at different levels [25]. It has been shown that autophagy is implicated in the development of different immune cell populations [26], in antigen presentation [27] or in adaptive immunity [28]. Additionally, autophagy contributes to the control of innate immune signaling, participating in the finely tuned balance between activated and repressed immune responses [29]. In fact, autophagy dysregulation can alter this equilibrium, leading to chronic inflammatory diseases [30]. This could explain why mutations of autophagy-related genes have been associated with several autoimmune disorders, including systemic lupus erythematosus (SLE), different types of sclerosis or rheumatoid arthritis [31]. Well-studied examples of autophagy dysregulation in uncontrolled inflammatory responses are inflammatory bowel diseases (IBDs), particularly Crohn’s disease [32]. Autophagy is also an important defense barrier against infection, as it contributes to the degradation of intracellular pathogens (i.e., virus or bacteria) [33]. However, some of these infective agents have acquired molecular mechanisms to evade and use the autophagic machinery for their benefit, which aggravates infection in some cases [34].

Undoubtedly, one of the clinical contexts in which the link between deficient autophagy and pathology is more firmly-established is neurodegeneration [35]. In this regard, the inability to clear the accumulation of aggregation-prone misfolded proteins (including β-amyloid, huntingtin or α-synuclein) hampers cell viability, leading to the progressive loss of central nervous system function. This is the case of disorders such as Alzheimer’s, Huntington’s, or Parkinson’s diseases, with some of them also showing problems in mitophagy, the selective autophagic degradation of mitochondria [36]. Development of amyotrophic lateral sclerosis (ALS) has also been associated with the accumulation of different protein aggregates or defective mitochondrial clearance, and pathogenic variants in components of the autophagic pathway have been described in patients [37]. Additional neurological disorders that have been linked to autophagy failure are spastic paraplegias, beta-propeller protein-associated neurodegeneration or Charcot–Marie–Tooth diseases, where autophagosome maturation, transport and/or its fusion with the lysosome are blocked [38].

The musculoskeletal system also requires autophagy to maintain its homeostasis, and autophagy dysregulation has also been associated with musculoskeletal pathologies. For example, different myopathies have been termed autophagic myopathies, as most of them show blocked autophagy flux and accumulation of autophagic vacuoles [39]. In bone tissue, autophagy plays an important role in controlling the balance between bone resorption and formation. Consistently, alterations in the autophagic pathway have been found in different diseases caused by the perturbation of bone physiology, such as Paget’s disease, osteopetrosis or osteoporosis [40]. Moreover, chondrocytes of the cartilage are also more susceptible to cell death when autophagy is disrupted, leading to osteoarthritis [41]. 

Remarkably, both neurological and musculoskeletal alterations are common features observed in lysosomal storage diseases (LSDs). LSDs are rare metabolic disorders caused by alterations in genes that are required for lysosomal-mediated degradation [42]. These pathogenic variants often result in the accumulation of specific undegraded substrates in the lumen of this organelle, hindering lysosomal function [43]. The identity of the unprocessed molecule (sphingolipids, glycogen, glycosaminoglycans, etc.) is the base of the classification of LSDs. For example, mucopolysaccharidoses (MPSs) are mainly caused by mutations in specific lysosomal genes that contribute to the degradation of glycosaminoglycans (GAGs), while Danon disease and Pompe disease are characterized by the intralysosomal accumulation of glycogen. Deficiency in the catabolism of glucocerebrosides causes Gaucher disease and failure to degrade globotriaosylceramide results in Fabry disease. Niemann–Pick type C disease is caused by the accumulation of unesterified cholesterol in several organs, while cystinosis is characterized by the accumulation of the amino acid cystine. Other important LSDs are galactosialidosis (with sialyloligosaccharide accumulation), fucosidosis (witch lysosomal aggregation of different molecules containing fucose moieties), mannosidosis (characterized by a deficiency in the degradation of mannose-rich oligosaccharides) and sialic acid storage diseases. Given the intricate connection between autophagy and lysosomal activity, it is not surprising that defects on any of them have an impact on the other one [44,45].

## 3. Relevant Variants on Autophagy-Related Genes

As shown in Figure 2, autophagic degradation involves different sequential stages, which operate from the regulation of autophagosome biogenesis to the last steps of autophagosome cargo degradation and recycling: (1) autophagy initiation, (2) membrane nucleation, (3) pre-autophagosomal membrane expansion, (4) autophagosome fusion with lysosomes and (5) degradation and efflux of basic components [46,47,48]. Each of these steps requires the coordinated temporal and spatial activation of several molecular components, namely the ULK1/2 kinase protein complex; the class III phosphatidylinositol 3-kinase (PI3KC3) protein complexes; phosphatidylinositol 3-phosphate (PI(3)P)-binding proteins and the ATG9-containing membranes; the ATG12 and ATG8 UBL conjugation systems; the selective autophagy receptors and the factors involved in autophagosome-lysosome fusion. Interestingly, pathogenic gene variants have already been described for all these different groups of effectors. 

### 3.1. The ULK1/2 Kinase Complex

The members of the ULK (Unc-51-like kinase) family of proteins are the orthologues of the yeast Atg1, a serine/threonine protein kinase essential for autophagy initiation (Figure 3). In human cells, there are five ULK proteins (ULK1, ULK2, ULK3, ULK4 and STK36) although among them, only ULK1 and ULK2 are involved in autophagy [48]. In living cells, ULK1 or ULK2 are part of a protein complex with at least ATG13, ATG101 and FIP200 (family-interacting protein of 200kDa, also known as RB1CC1). This complex is responsible for driving autophagy initiation upon autophagy-inducing stimuli [49]. When active, the ULK1/2 complex translocates to autophagosome formation sites and regulates the recruitment and activation of the class III phosphatidylinositol 3-kinase (PI3KC3) complex, which in turn will generate phosphatidylinositol 3-phosphate (PI(3)P), a signaling molecule that recruits other downstream factors involved in autophagosome biogenesis. Moreover, the ULK1/2 protein complex carries other different autophagy-related functions, such as ATG9-vesicle recruitment or regulation of ATG4B activity, and contributes to regulate mitophagy and degradation of protein aggregates [50]. Due to its importance in autophagy initiation, this protein complex is regulated by a variety of post-translational modifications, such as acetylation, ubiquitin conjugation or phosphorylation by protein kinases. Among these, adenosine monophosphate-activated protein kinase (AMPK) and mechanistic/mammalian target of rapamycin (mTOR) are the most relevant, connecting ULK1/2 complex activity to the nutritional and energetic status of the cell [51,52].

Several pathogenic variants of the ULK1/2 kinase complex have been identified (Table 1 and Appendix A). For example, SNPs in ULK1 have been associated with Crohn’s disease susceptibility and clinical outcomes in different populations [53,54], supporting previous evidence of autophagy alterations in inflammatory bowel diseases. Additional variants of *ULK1* have shown strong associations with tuberculosis [55,56] and also with a specific type of rheumatoid arthritis termed ankylosing spondylitis [57], further linking ULK1 activity with the immune system. SNPs in *ULK2* have only been linked to asparaginase-associated pancreatitis to date [58]. Also related with the immune system is an ATG13 variant that may be associated with selective immunoglobulin A deficiency (IgAD), although it is not clear if this polymorphism affects *ATG13* or *AMBRA1* (which is another gene involved in autophagy) [59]. Additionally, altered ATG13 activity may also be involved in chemotherapy-induced cardiotoxicity in triple-negative breast cancer patients [60]. Regarding *FIP200* only one pathogenic SNP has been documented, which predicts hypertension after metastatic colorectal cancer treatment [61]. In synthesis, polymorphisms in members of the ULK1/2 kinase complex have been associated with a variety of pathologies, some of them related to immune system dysfunction. It is also remarkable that seven different pathogenic SNPs have already been identified in the *ULK1* gene (Figure 2). It is also noteworthy that different variants of this gene have been linked to a determined pathology (such as Crohn’s disease or tuberculosis).

### 3.2. The Class III Phosphatidylinositol 3-Kinase (PI3KC3) Complexes

After being activated at the assembling site, the ULK1/2 kinase complex acts as a scaffold for the PI3KC3 complex, whose activity is essential for the nucleation of the pre-autophagosomal membranes by generating PI(3)P, an essential signal for autophagosome formation, which will in turn recruit additional downstream factors involved in autophagosome biogenesis [49] (Figure 4). This complex is formed by Beclin 1, VPS34/PIK3C3, VPS15/p150/PIK3R4 and Barkor/ATG14L [62]. The most important event for PI3KC3 regulation is the sequestration of Beclin 1 by BCL-2, which limits its ability to bind PI3KC3 complexes, resulting in autophagy inhibition [63]. Conversely, AMBRA1 can also bind to Beclin 1 (and other autophagy-related proteins), increasing PI3KC3 complex activity and thus supporting autophagosome formation [64]. Interestingly, there is an additional version of the PI3KC3 complex (often called PI3KC3-C2) in human cells, which is also involved in autophagy regulation. PI3KC3-C2 contains UVRAG instead of ATG14 and is involved in regulating the fusion of autophagosomes to lysosomes [65]. The activity of this complex can be repressed by binding of the negative regulator Rubicon/KIAA0226, which results in the inhibition of autophagosome/lysosome fusion [66,67]. Conversely, its activity can be enhanced by binding of the protein associated with UVRAG as autophagy enhancer (Pacer), which increases autophagic degradation [68].

Despite the relevance of Beclin 1 in autophagy, only two SNPs found in the *BECN1* gene have so far been associated with diabetes [69] and Machado–Joseph disease [70], a neurodegenerative disorder characterized by progressive cerebellar ataxia (Table 2 and Appendix A). On the other hand, polymorphisms on *VPS34/PIK3C3* do correlate with increased cancer risk, specifically in pancreatic adenocarcinoma [71] and esophageal squamous cell carcinoma [72,73]. A third variant has been linked to gastric cardia adenocarcinoma, although it seems to be related to non-autophagical functions of VPS34 in the control of telomere length [74]. Another single nucleotide change on the promoter of *VPS34* has been associated with both systemic lupus erythematosus [75] and with bipolar disorder and schizophrenia [76]. Interestingly, several variants of *AMBRA1* have also been linked to schizophrenia [77], as well as to diverse forms of autism [78]. A polymorphism in *ATG14* has been associated with testicular germ cell tumors [79], while different *UVRAG* alleles have been linked to a less efficient treatment response in multiple sclerosis [80], susceptibility to rheumatoid arthritis [81] and non-segmental vitiligo [82]. Altogether, these findings show that genetic variants in components of the PI3KC3 complexes are linked to a variety of pathologies, including cancer, autoimmune or neurological disorders. Among all of them, *VPS34* and *UVRAG* seem to be more sensitive to nucleotide changes, with *AMBRA1* (often associated with PI3KC3 complexes) also accumulating several pathogenic polymorphisms (Figure 2).

### 3.3. PI(3)P-Binding Proteins and the ATG9-Containing Membranes

Production of PI(3)P by the PI3KC3 complex acts as a signal for the recruitment of autophagy-related proteins able to bind this phospholipid. Among these, the most characterized are DFCP1, WIPI proteins and ALFY [83]. DFCP1 recruitment to autophagosome formation sites occurs shortly after PI(3)P generation by PI3KC3. There, it labels the omegasome, a platform for the formation of autophagosomes that originate from the ER-associated membranes. DFCP1 colocalizes with LC3 and other essential proteins for autophagosome biogenesis. However, despite DFCP1 being often used as a marker for the omegasome, its depletion has no major effect on autophagy [84]. WIPI proteins are also recruited by the presence of PI(3)P at the pre-autophagosomal membranes, where they contribute to autophagosome formation by recruiting other autophagy essential proteins for this process [85] (Figure 5). In human cells, there are four WIPI proteins (WIPI 1-4). WIPI1 and WIPI2 proteins participate in the expansion of the autophagosomal membrane, with WIPI2 being responsible for recruiting the ATG5–ATG12/ATG16 complex to nascent autophagosomes [86]. In contrast, WIPI3 and WIPI4 interact with upstream regulators, such as the ULK1/2 complex and AMPK-activated TSC complex, coupling PI(3)P generation to the activity of upstream regulatory signaling [87]. Moreover, WIPI4 plays an important role in controlling the growth and size of autophagosomes through its interaction with ATG2 proteins. In yeast, Atg2 acts as a lipid transfer protein that supplies phospholipids to autophagosomal membranes [88]. In mammals, there are two Atg2 orthologues, ATG2A and ATG2B, that can translocate lipids from other membranes to nascent autophagosomes, forming a complex with WIPI4 that acts as a membrane tether with lipid transfer activity [89]. ALFY (autophagy-FYVE-linked protein) is also a PI(3)P-binding protein, which is recruited to nascent autophagosomal membranes close to protein aggregates. Although ALFY is not essential for autophagosome biogenesis during starvation-induced autophagy, it is required for protein aggregate autophagic degradation [90]. The only transmembrane proteins directly involved in autophagosome biogenesis are those from the ATG9 family, composed by ATG9A and ATG9B. ATG9 proteins assist with lipid transport and transferring to nascent autophagosomal membranes [49,91] (Figure 5). Concurrently with PI3KC3 complex activation, the ULK1 kinase complex also mediates the recruitment of vesicles containing either ATG9A or ATG9B (depending on the tissue or cell type). These transmembrane proteins drive the aforementioned vesicles to the assembling site so they can be used as membrane sources, contributing to the elongation of the autophagosomal membrane [83]. WIPI2 is also involved in the regulation of ATG9 traffic, as WIPI2 depletion hampers ATG9 dynamics and results in the accumulation of ATG9 pre-autophagosomal structures due to the inhibition of ATG9 retrieval to Golgi complex membranes [92]. 

To date, only one pathogenic SNP on *DFCP1* has been found, which is linked to tuberculosis resistance [93]. In contrast, nucleotide changes on *ALFY* were identified in patients with microcephaly [94], as well as in oropharynx cancer [95]. *WIPI4* pathogenic SNPs have been shown to cause neurological disorders, such as neurodegeneration with brain iron accumulation (NBIA) and Rett syndrome [96,97,98], while polymorphisms on *WIPI2* and *WIPI3* have been associated with osteoporosis [99] and a neurodevelopmental syndrome [100]. Two variants of *ATG2A* have been linked with both granuloma formation in Crohn’s disease and hyperuricemia [101,102]. Interestingly, the equivalent SNP in *ATG2B* increases susceptibility to neck squamous cell carcinoma in pharyngeal cancer [103] and correlates with both progression and recurrence of bladder cancer after treatment with bacillus Calmette–Guérin intravesical instillation [104]. Finally, a polymorphism associated with coronary artery disease has been identified on *ATG9B* [105] (Table 3 and Appendix A). In summary, pathogenic SNPs that have been described on lipid transfer mediators are linked to pathologies of different origins. In contrast, pathogenic SNPs on genes coding for PI(3)P-binding proteins are mostly associated with neurological disorders. It is remarkable that distinct pathogenic polymorphisms related to very different diseases have been described for *WIPI4* (Figure 2).

### 3.4. The ATG12 and ATG8 Conjugation Systems

Two ubiquitin-like (UBL) conjugation systems are essential for autophagosome formation and autophagic cargo degradation (Figure 6). These UBL systems cooperate to drive the expansion of the nascent autophagosomal membrane, with the final product of the first system being the responsible enzyme for the last reaction in the second one [106]. The first UBL system, the ATG12 system, mediates the activation, transfer, and covalent conjugation of ATG12 to ATG5, a process that requires the sequential activities of the E1-like enzyme ATG7 and the E2-like enzyme ATG10. Once formed, two molecules of the ATG12–ATG5 conjugate interact with an ATG16 dimer, resulting in the final ATG16 complex. This complex is recruited to nascent autophagosomes by the actions of WIPI1 and WIPI2 proteins [86]. After being recruited, ATG16 keeps the ATG5–ATG12 conjugate at the pre-autophagosomal membrane, where it acts as an E3-ligase for the ATG8 UBL system, catalyzing ATG8 covalent conjugation [107]. It is interesting that although there are two ATG16 homologues (ATG16L1-2) in human cells, only ATG16L1 seems to play a prominent role in autophagy [108,109].

The second UBL system, the ATG8 system, mediates the conjugation of ATG8 molecules to phosphatidyl-ethanolamine (PE) phospholipids at the pre-autophagosomal membrane (Figure 6). In humans, there are six ATG8 proteins, grouped into two subfamilies: the MAP1-LC3 subfamily (including MAP1-LC3A, MAP1-LC3B and MAP1-LC3C) and the GABARAP subfamily (including GABARAP, GABARAPL1/ATG8L and GABARAPL2/GATE-16). All these proteins can be found either free or in their lipidated (PE-conjugated) form in human cells [110]. In normal conditions, the unconjugated ATG8 forms are mostly cytosolic. In contrast, ATG8 PE-conjugated forms are mainly associated with the inner and outer membranes of the autophagosome. ATG8 conjugation first requires the activity of ATG4 proteases (ATG4A-D, also called “autophagins” in humans) [111]. ATG4s cleave and activate ATG8 proteins, leaving a glycine residue at the carboxyl terminus, which is essential for ATG8 lipidation. ATG7 (E1-like enzyme), ATG3 (E2-like enzyme) and the ATG12-ATG5–ATG16L1 multimeric complex (E3-like enzyme) are necessary for ATG8 conjugation to PE. This conjugation is, in turn, essential for the expansion of the pre-autophagosomal membrane, autophagosome maturation and degradation of autophagic cargo [106]. Once ATG8s’ presence at the autophagosomal/autolysosomal membranes is no longer required, they can be deconjugated from PE by the action of ATG4s. In yeast, Atg8 deconjugation is important to keep an available pool of ready-to-use Atg8 in the cytosol and required for efficient autophagosome biogenesis and maturation [112,113]. 

Interestingly, most of the pathogenic polymorphisms that have been already identified on autophagy-related genes are located on the genomic sequence of members of the ATG12 conjugation system (Table 4 and Appendix A). In fact, variants affecting different genes from this system are associated with the same pathology. For example, pathogenic alleles of *ATG5*, *ATG10*, *ATG12* and *ATG16L1* have been all linked to changes in susceptibility or treatment efficiency in neck squamous cell cancer [103,114,115,116], hepatocellular carcinoma [117] and lung adenocarcinoma [118] and others have been found to also affect development of other types of cancer, such as melanoma [119], brain metastases in patients with non-small lung cancer [120] or breast cancer [121,122,123,124]. Moreover, there is also an association between SNPs in *ATG5* and *ATG7* with clear cell renal cell carcinoma [125] and variants of *ATG5* and *ATG10* have been linked to non-small cell lung cancer [126,127]. Additional connections between changes on these genes and cancer are those of *ATG5* in multiple myeloma [128] or non-medullary thyroid cancer [129]. Different *ATG16L1* SNPs have been linked to cell-derived thyroid carcinoma [130], colorectal cancer [131], gastric cancer [132] or prostate cancer [133]. Moreover, several polymorphisms on the ATG12 conjugation system have also been associated with other pathologies besides cancer. In fact, SNPs in *ATG5* and *ATG7* are linked to the development of neurological disorders such as cerebral palsy (both *ATG5* and *ATG7*) [134,135], Huntington’s disease (*ATG7*) [136,137], and Parkinson’s disease or spinocerebellar ataxia (*ATG5*) [138,139]. Additionally, nucleotide polymorphisms on *ATG5*, *ATG10* and *ATG12* have been related to pneumoconiosis in a population of coal workers [140], whereas other SNPs in *ATG7* have been associated with ischemic stroke [141].

Perhaps the most studied SNP on an autophagy-related gene is rs2241880 on *ATG16L1*, resulting in a threonine-to-alanine substitution at amino acid position 300 (T300A). Its link to inflammatory bowel disease, first described by Hampe and collaborators [142], has been extensively confirmed in different populations by an immeasurable list of studies, impossible to entirely cite in this review. Polymorphisms on other proteins with autophagy-related functions have also been associated with inflammatory bowel diseases. For example, variants of *IRGM*, an important effector that links the autophagy molecular core to innate immunity receptors [143], are well-known risk alleles in pathologies such as Crohn’s disease [144]. Interestingly, two SNPs in *ATG5* correlate to a positive response to Crohn’s disease therapy [145]. Moreover, variants of *ATG16L1* and other effectors of the ATG12 conjugation system have also been linked to other inflammatory disorders, autoimmune diseases or other complications related to the immune system. In this regard, polymorphisms on *ATG5*, *ATG10* and *ATG16L1* have been associated with Paget’s disease of the bone [146], and SNPs in *ATG5* and *ATG7* have been extensively studied in the context of systemic lupus erythematosus [147,148,149,150,151,152,153]. In addition, several polymorphisms on *ATG5* may play a role in the pathogenesis of other autoimmune disorders such as Behçet’s disease [154], neuromyelitis optica [155] and systemic sclerosis [156,157]. Additional inflammatory alterations linked to variants of *ATG5* are aplastic anemia [158] or asthma [159,160] as well as complications related to infections, including chronic Q fever [161], sepsis [162] or Hepatitis B [163,164]. Similarly, SNPs in *ATG16L1* are also associated with *Helicobacter pylori* infection and related gastric cancer [165,166,167] or skin conditions such as palmoplantar pustulosis [168] and psoriasis [169]. Finally, a variant of *ATG10* has been associated with Vogt–Koyanagi–Harada disease, which is characterized by an autoimmune response against melanin-producing cells [154].

Fewer clinically relevant variants have been identified in genes encoding members of the ATG4 and ATG8 protein families (Table 4 and Appendix A), perhaps because of their marked redundancy. Different SNPs in *ATG4A* have been associated with kidney, cervical and lung cancer [125,170,171]. Meanwhile, variants of *ATG4B* may be present in patients with obesity [172] or atherosclerosis [173]. Polymorphisms on *ATG4C* have also been linked to clear cell renal cell carcinoma [125], as well as an increase in susceptibility to Kashin–Beck disease, a osteochondropathy characterized by chondrocyte death and altered autophagy in growth plate and articular cartilage [174]. Nucleotide changes on *ATG4A* and *ATG4D* genes can also lead to the formation of granulomas during Crohn’s disease [101]. As for their substrates, the ATG8 proteins, only three gene variants have been associated with a pathology so far: two polymorphisms on *MAP1LC3A* that may contribute to progression of chronic Q fever [161] or coronary artery disease [175], and two SNPs that alter *MAP1LC3B* expression and correlate to myopia [176] and increased susceptibility to systemic lupus erythematosus [177]. To date, no pathogenic SNPs have been identified either in *MAP1LC3C*, *GABARAP, GABARAPL1, GABARAPL2* nor in the gene encoding the E2-like enzyme ATG3.

In summary, alterations in autophagy UBL systems have been extensively linked to disease. This is shown not only by the high number of SNPs found in genes of these systems, but also by the total number of diseases to which they are associated (Figure 2). Specifically, genes such as *ATG16L1*, *ATG5*, *ATG10* or *ATG7*, accumulate multiple SNPs linked to pathologies that include cancer, neurological disorders, or inflammatory bowel diseases.

### 3.5. Selective Autophagy Receptors

Autophagic degradation can be either bulk or selective. The last one requires the action of the so-called selective autophagy receptors (SARs), which mediate the recognition and engulfment of specific cargo in autophagosomes (Figure 7). Specifically, these adapters can simultaneously bind both to the target molecules and to the ATG8 proteins conjugated on the concave side of the autophagosomal membrane [178]. The identification and study of these SARs is likely one of the most exciting, fast-paced fields of autophagy research, as autophagy adapters show specificity to a wide variety of substrates. The most characterized mammalian SARs are those that bind ubiquitin molecules [179]. These molecules label a great variety of autophagic substrates, from protein aggregates to damaged mitochondria or intracellular pathogens. The most studied ubiquitin-binding SAR is p62/SQSTM1 [180], a multifunctional protein important for protein aggregate degradation (aggrephagy), mitophagy and the engulfment of intracellular pathogens by autophagosomes (xenophagy). Similarly, NBR1 is also involved in aggrephagy and acts synergically with p62/SQSTM1 for the degradation of ubiquitin-decorated protein aggregates [181]. OPTN is not only involved in aggrephagy [182] but also in mitophagy [183] and xenophagy [184]. Similarly, NDP52 is involved both in mitophagy [185] and xenophagy [186], as TAX1BP1 is [183,187]. TOLLIP has also been involved in mutant huntingtin-selective degradation, although it is less studied than other ubiquitin-binding SARs [188]. Apart from this group of SARs, many other proteins, each of them localized in a specific cellular structure/organelle, interact with ATG8s and act as specific SARs for their contained subcellular structures [189]. This is the case for BNIP3, BNIP3L/NIX, FUNDC1, FKBP8, PHB2 or NIPSNAP1/2, which are specific mitophagy SARs [190,191,192], or that of FAM134B, Sec62, RTN3, CCPG1, ATL3 and TEX264, which are specific SARs for ER-selective autophagy [193]. Apart from these, multiple other specific autophagy receptors for other selective autophagic processes are being constantly identified. This is the case of ATGL and HSL for lipophagy [194], STBD1 for glycophagy [195], NUFIP1 for ribophagy [196], or NCOA4 for ferritin degradation [197,198]. Certainly, new specific receptors for orphan-selective autophagic processes, such as zymophagy or granulophagy will be identified in the future.

Several studies have described important pathogenic SNPs in the genes of selective autophagy receptors (Table 5 and Appendix A). Nucleotide changes on *SQSTM1*, for example, may contribute to the origin of neurological alterations [199], as well as being implicated in the pathogenesis of amyotrophic lateral sclerosis [200,201], dementia [202], apraxia of speech [203], myopathy [204] and Paget’s disease of the bone [205,206]. Additionally, *NDP52* and *NBR1* variants determine susceptibility to Crohn’s disease [207] and Brooke–Spiegler syndrome [208], a rare condition where tumors form from skin structures. Meanwhile, polymorphisms on *OPTN* also show clinical relevance, as they are associated with amyotrophic lateral sclerosis [209], Paget’s disease of the bone [210,211] and primary open-angle glaucoma [212]. Interestingly, SNPs in *TOLLIP* have been frequently linked to interstitial lung diseases [213] and different types of infections, including leishmaniasis [214], leprosy [215,216], malaria [217], tuberculosis [218] and septicemia [219]. Pathogenic alleles of *TAX1BP1* have been identified in unrelated alterations such as oral cavity cancer [220] and hypospadias [221]. Variants of specific mitophagy adapters have been identified in patients with depression (in the case of *BNIP3*) [222], schizophrenia or impaired cognition (BNIP3L) [223], bisphosphonate-associated osteonecrosis of the jaw (PHB2) [224], and breast cancer (NIPSNAP1) [123]. Regarding reticulophagy receptors, patients with hereditary sensory autonomic neuropathy have SNPs in either *FAM134B* [225,226] or *ATL3* [227,228], and a single nucleotide change on *RTN3* increases the susceptibility to complications after malaria [229,230]. Changes on genes encoding lipophagy receptors ATGL and HSL are unsurprisingly associated with neutral lipid storage disease with myopathy [231] and familial partial lipodystrophy [232,233]. Additional links between polymorphisms on SARs genes and pathologies are those in *STBD1* for Parkinson’s disease [234,235], *NUFIP1* for asthma [236], and *NCOA4* for cancer [237,238]. Finally, a high number of pathogenic variants in the gene encoding huntingtin protein (HTT) are the cause for Huntington’s disease [239]. Although it might not be considered a bona fide selective autophagy receptor, HTT can act as a scaffold for selective autophagy and interacts with ULK1, GABARAP and p62/SQSTM1 [193,240,241]. Taken together, these publications show that dysregulation of autophagy receptor function plays a role in the pathogenesis of a wide variety of diseases, with most pathogenic variants being identified in *SQSTM1, TOLLIP* and *HTT* (Figure 2).

### 3.6. Cellular Machineries Involved in Autophagosome-Lysosome Fusion

Once fully formed, autophagosomes move along microtubules depending on the actions of the minus-end-directed motor protein dynein and a plus-end-directed motor kinesin/FYCO1 [242]. This bidirectional transport leads to autophagosome clustering around the perinuclear area, where they eventually fuse with lysosomes [243]. In fact, disruption of either dynein or KIF5B, the heavy chain of kinesin-1, impairs autophagosome/lysosome fusion, blocking autophagic degradation [244,245]. Once mature autophagosomes and lysosomes encounter, lysosomal and outer autophagosomal membranes fuse forming a new organelle called an autolysosome, in which degradation of autophagic cargo occurs [246]. This fusion requires the coordination of SNAREs, small GTPases, tethering factors, and other proteins [247]. The SNARE proteins involved in autophagosome/lysosome fusion are the Q-SNAREs STX17 and SNAP29 and the R-SNARE YKT6, all present at the autophagosomal membrane. At the lysosomal membrane, R-SNAREs such as VAMP8 or VAMP7 and Q-SNARE STX7 have been reported to interact with autophagosomal SNAREs to mediate membrane fusion. Rab GTPases also play a major role in this process, recruiting other proteins that act coordinately to enhance the efficiency and specificity of fusion [248,249]. In this context, Rab7 is likely the most important Rab protein, as it has been reported to recruit tethering factors, including EPG5, PLEKHM1, and VPS33A and VPS41 from the HOPS complex [250], which all promote the assembly of trans-SNARE complexes for fusion [251,252]. A similar role for the Rab2 protein has also been proposed in autophagy [253,254] and Rab33b has been shown to recruit the ATG16L1 complex to pre-autophagosomal membranes [255]. Moreover, ATG14L has also been shown to act in this process stabilizing the STX17–SNAP29 complex to promote autophagosome/lysosome fusion [256]. Finally, other tethering factors, such as GRS2/GRASP55 [257] or BIRC6/BRUCE [258] have been reported to play a role in this process.

To become autolysosomes, autophagosomes may either fuse directly with lysosomes or fuse their external membrane with endosomes and become an organelle called amphisome, which will eventually fuse with lysosomes [259]. This event is sometimes required for efficient autolysosome formation and disruptions in autophagosome/endosome fusion often result in autophagosome accumulation and autophagic degradation blockage [260]. Members of the ESCRT families of proteins have been shown to be required for autophagosome/endosome fusion, and thus for adequate autolysosome formation. In this regard, the ESCRT-associated AAA-ATPase VPS4B/SKD1 has been shown to be required for efficient autophagosome clearance [261]. Depletion of ESCRT-0 HRS/HGS also leads to impairment in autophagosome maturation and fusion with lysosomes [262]. Similarly, depletion of CHMP4B/SNF7-2/VPS32-2 (ESCRT-III) or the expression of a mutant form of CHMP2B causes an accumulation of autophagosomes [263]. Other proteins that have been suggested to be involved in endocytic transport and autophagy are C9orf72 [264], ZFYVE26, SPG11 [265], and Rab33B [266].

Altogether, the coordinated activity of all these proteins results in autolysosome formation, which is the last step preceding autophagosome cargo degradation (Figure 8). Table 6 shows polymorphisms on the genes encoding these effectors that are connected to diseases (with additional references collected in Appendix A). Regarding the genes encoding motor proteins, SNPs in *FYCO1* and *KIF5B* have been associated with cataracts [267] and bipolar disorder [268], respectively. As for genes encoding SNARE proteins, several links with alterations have been established: *STX7* and neuronal heterotopia [269], *STX17* and alopecia [270], *YKT6* and diabetes or birth weight [271,272], and *SNAP29* and a neurocutaneous condition termed Cednik syndrome [273]. Variants of another SNARE, *VAMP8*, are present in patients with coronary artery disease [274], cerebrovascular accident [275], tuberculosis [276] or prostate cancer [277]. An SNP in *Rab7* has been identified in patients with Charcot–Marie–Tooth disease type 2B [278,279,280], while SNPs in *Rab33B* are associated with Smith–McCort osteochondrodysplasia [281,282,283]. As for changes on genes from members of HOPS complex, a variant of *VPS33A* has been linked to a new type of mucopolysaccharidosis [284], with a specific allele of *VPS41* being implicated in major depressive disorder [285]. Additionally, SNPs in *PLEKHM1* have been involved in several diseases, including osteopetrosis [286], Parkinson’s disease [287], ovarian cancer [288,289], depression [290], and alopecia [291]. Some pathogenic variants on *EPG5* are responsible for Vici syndrome, a congenital multisystem disorder [292], while others have also been linked to Alzheimer’s disease [293] or depression [294]. Only one polymorphism of *BIRC6* has been described in a disease, specifically glaucoma [295]. Variants on components of the ESCRT-III complex component are associated with a wide range of disorders. For example, SNPs in *CHMP2B* are linked to neuroblastoma [296], frontotemporal dementia [297,298] or ALS [299] and those in *CHMP4B* have been associated with cataracts [300], dysphagia [301] or diabetes mellitus [302]. Polymorphisms in *C9orf72* are also associated with frontotemporal dementia and ALS [264,303], while one SNP in *HGS* correlates with age-related macular degeneration [304]. Finally, variants of *ZFYVE26* [305] and *SPG11* [306] have been extensively analyzed in patients with spastic paraplegia, while SNPs in *ZFYVE26* have been also linked to ALS [307] and breast cancer [308]. All in all, polymorphisms on genes encoding proteins involved in autophagosome-lysosome fusion often result in the development of different diseases, with dementia, ALS and paraplegia being the most frequent ones. It is remarkable that a large number of clinically relevant SNPs have been identified in the genes encoding EPG5, SPG11 and ZFYVE26 (Figure 2).

Once the autolysosome is formed, the inner membrane and the internal content of the original autophagosome are degraded by the action of acidic hydrolases (Figure 9). After that, the resulting new biomolecules (nucleotides, lipids, amino acids, etc.) return to the cytoplasm by the action of permeases and other transporters. Therefore, the lysosome becomes an essential player in autophagy. Disruption of lysosomal activity impacts autophagic degradation, leading to the accumulation of autophagosomes and/or autolysosomes, which physically stresses the cell while undesired cytoplasmic components accumulate without being degraded. In fact, alterations in the balance of lysosomal lipids can hinder autophagy, either by blocking autophagosome-lysosome fusion (which is the case in multiple sulfatase deficiency (MSD) or in mucopolysaccharidosis type IIIA [309,310]), impeding autophagosomal closure (by, for example, the accumulation of sphingomyelin in Niemann–Pick type A and B [311]), impairing lysosomal proteolysis (shown in Niemann–Pick disease type C [312]) or resulting in lysosomal permeabilization (as it has been described in Niemann–Pick disease type A [313]).

Classic examples of human pathologies caused by alterations in lysosomal function are LSDs. These pathologies are originated by variants that alter the activity of either specific lysosomal proteins (including hydrolases, transferases, membrane proteins, activators or transporters) or non-lysosomal ones that are required for the lysosomal-mediated degradation of different molecules. Thus, glycosaminoglycans are accumulated in different types of mucopolysaccharidoses. In fact, a long list of diseases are originated by pathogenic SNPs in the genes involved in their degradation, such as *ARSB* (MPS VI or Maroteaux–Lamy syndrome) [314], *GALNS* (MPS IVA or Morchio A syndrome) [315], *GLB1* (MPS IVB or Morchio B syndrome) [316,317,318], *GNS* (MPS IIID or Sanfilippo syndrome type D) [319] *GUSB* (MPS VII or Sly syndrome) [320], *HGSNAT* (MPS IIIC or Sanfilippo syndrome type C) [319], *HYAL1* (MPS IX) [321], *IDS* (MPS II or Hunter syndrome) [322], *IDUA* (different subtypes of MPS I and II) [323], *NAGLU* (MPS IIIB or Sanfilippo syndrome type B) [319] or *SGSH* MPS IIIA or Sanfilippo syndrome type A) [319]. One of the most well-studied pathologies caused by lysosomal deficiency is Danon disease, which has been extensively associated with LAMP2 deficiency [324]. Another well-documented glycogen storage disorder, Pompe disease, is caused by mutations in *GAA* [325], which has also been implicated in Friedreich ataxia [326] and hypochondrogenesis [327]. Deficiency of the lysosomal hydrolase β-glucocerebrosidase (GCase), encoded by *GBA*, causes Gaucher disease [328], and a polymorphism on this gene has also been linked to Parkinson’s disease [329], whereas several pathogenic SNPs in *GLA*, the gene encoding the lysosomal enzyme α-galactosidase A (a-Gal A), are associated with Fabry disease [330]. Polymorphisms in *NPC1* and *NPC2* genes results in Niemann–Pick type C disease [331], while changes in *CTNS* have been associated with cystinosis [332]. Different types of neuronal ceroid lipofuscinoses can be caused by mutations on different genes, such as *GRN*, several *CLNs*, *CTSD*, *CTSF*, *DNAJC5* or *MFSD8* [333]. Finally, pathogenic SNPs on *CTSA*, *FUCA1*, *SLC17A5*, and *MANBA* or *MAN2B1* lead to galactosialidosis [334], fucosidosis [335], sialic acid storage diseases [336] or mannosidosis [337,338], respectively. All these clinically relevant SNPs are shown in Table 7, with additional references collected in Appendix A. It is remarkable that numerous SNPs on a given lysosomal gene are only associated with just one or two pathologies (*GLB1* being the single exception) (Figure 2). This contrasts with the case of the genes encoding proteins involved in autophagosome biogenesis, in which fewer variants are linked to a wider range of diseases.

## 4. Concluding Remarks

As the number of studies on autophagy research grows, it becomes clearer how essential this catabolic pathway is for cellular homeostasis and health. What was first inferred from the characterization of animal models deficient in autophagy it is now directly described in human pathology, with an increasing record of papers showing that autophagy dysregulation drives or sustains a wide range of disorders. The extensive list of autophagy-related SNPs collected in this review further reflects the relevance of autophagy and its effectors in human pathology. In this regard, several variants present in different populations can decisively affect the origin, development or prognosis of different diseases, remarking the importance of defining autophagy-related gene signatures in these disorders. Thus, the existence of variants on autophagy-related genes allows us to better understand the role of this route in pathophysiology.

Interestingly, in some cases, the very same SNPs have been described to be associated with different disorders either with a pathogenic or a protective effect. This is the case, for example, of rs2241880, the threonine-to-alanine substitution at amino acid position 300 of ATG16L1 (T300A) that has been intensively analyzed in the context of inflammatory bowel diseases. This variant has additionally been linked to different types of cancer or to Paget’s disease of the bone, and the same allele may be protective or pathogenic depending on the disorder and the population. This redundancy is also true for other genes, such as *ATG5* (rs510432, rs573775 or rs2245214) or *ATG10* (rs10514231, rs1864182 or rs1864183). Although identification of the same variant in different pathologies could be explained by the fact that already-studied SNPs are more likely to be analyzed again in the context of additional diseases, it nevertheless shows that alteration in the activity of autophagy proteins definitely contributes to the progression or repression of a given disorder. This bias could also explain why most of the autophagy-related polymorphisms that we have collected are located on genes of effectors mediating ATG12 conjugation (Table 4), as many researchers first focus on these proteins when addressing autophagy in a pathological context. Similarly, the analysis of SNPs in lysosomal genes has been favored by the direct link between mutations on these genes and lysosomal storage disorders (Table 7), entailing a fast, straightforward approach to find pathogenic variants in diseases. 

Finally, and although this review is focused on SNPs, it is worth mentioning that other less-frequent pathogenic variants of autophagy-related genes have also been described in the literature. For example, the monoallelic deletion of the 17q21 region, affecting the *BECN1* gene, is associated with breast, ovarian and prostate cancer [339,340]. Two variants of *WIPI4* (c.439 + 1G > T and c.1033_1034dupAA) were identified in patients with beta-propeller protein-associated neurodegeneration [98], while a third one is specifically linked to developmental and epileptic encephalopathy in patients with the same neurological disorder [341]. Sequence variants affecting the expression levels of *ATG12* (115842507G>T, 115842394C>T and 115841817_18del) [342] and *ATG7* (11313449G>A, 11313811T>C, 1313913G>A and 11314041G>A) [343] are present in patients with sporadic Parkinson’s disease. Other examples of less-common risk alleles are those of *SQSTM1* associated with frontotemporal dementia (c.1142C>T, K341V and K344E) [202,344] or muscle disorders such as sporadic inclusion body myositis (G194R) [204] and distal myopathies with rimmed vacuoles (p.G351_P388del and p.Glu389delinsAspLysTer) [345]. Although these and other minority variants have only been identified in few patients, their real allelic frequency in populations could be much higher, increasing their relevance and consolidating the clinical importance of pathogenic autophagy variants.

As a conclusion, it has to be considered that medicine is firmly progressing toward a personalized approach, given that patients respond differently to the same treatments. Although other factors are surely involved, the presence of SNPs plays a pivotal role in the specific response of a patient to a determined treatment. In this regard, autophagy-related polymorphisms are not only involved in the development of pathologies, but also may influence the effectiveness of a determined treatment. This additional clinical relevance of variants on autophagy-related genes has already been described in cancer [61,104,114,118,121], inflammatory bowel diseases [145,346,347] and others [80]. For this reason, the identification of new clinically significant SNPs is not only important in terms of disease prevention, but also to design new therapeutic approaches aimed at modulating autophagy for clinically relevant purposes. This review should be of great help in advancing to design new therapeutic strategies.

## Figures and Tables

**Figure 1 ijms-21-08196-f001:**
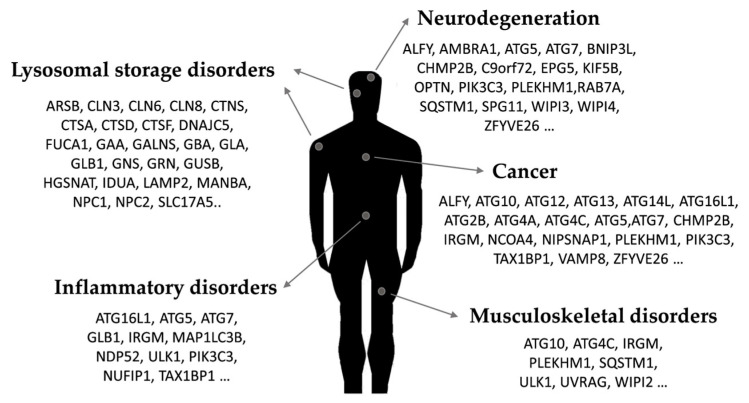
Representative links between autophagy-related proteins and human pathology.

**Figure 2 ijms-21-08196-f002:**
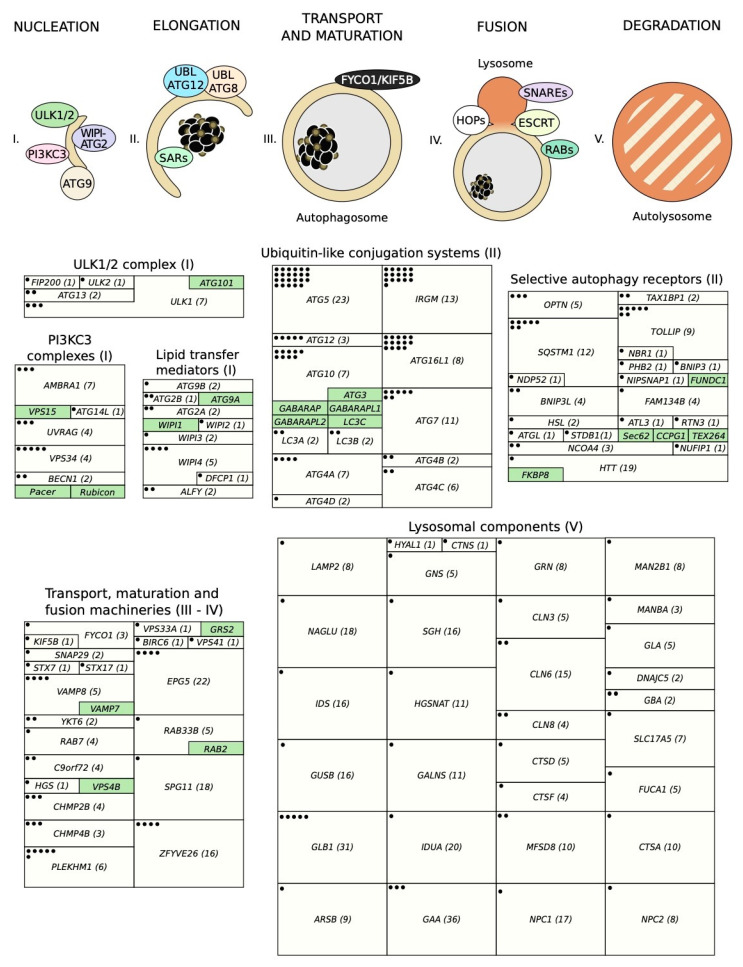
Schematic view of the incidence of clinically relevant single nucleotide polymorphisms (SNPs) in genes throughout the autophagosome/lysosome axis. Autophagy can be divided in different stages: (**I**) initiation and membrane nucleation, (II) membrane expansion, (**III**) autophagosome maturation and transport, (**IV**) autophagosome-lysosome fusion and (**V**) lysosomal degradation. Pathological variants have been found in genes involved in all of the steps. Boxes show the genes whose products participate in each of these stages. The size of the boxes is proportional to the number of pathological SNPs found for each of the genes depicted. The total number of pathological SNPs found in a given gene is depicted between brackets. Each different disease linked to a determined gene is represented by a dot. Green boxes contain genes without any clinically relevant SNPs identified to date. PI3KC3, class III phosphatidylinositol 3-kinase protein complexes; UBL, ubiquitin-like conjugation system; SARs, selective autophagy receptors; HOPS, homotypic fusion and protein sorting tethering complex; ESCRT, endosomal sorting complexes required for transport.

**Figure 3 ijms-21-08196-f003:**
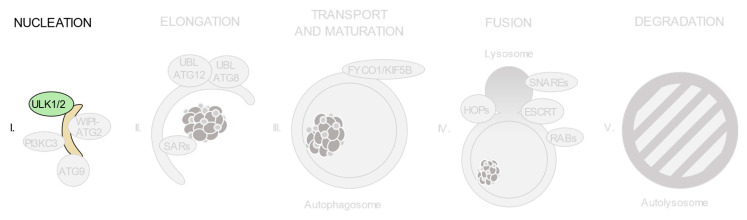
The ULK1/2 kinase complex participates in the nucleation of the pre-autophagosomal membrane.

**Figure 4 ijms-21-08196-f004:**
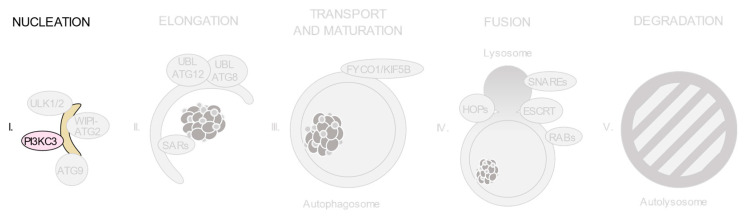
The phosphatidylinositol 3-kinase (PI3KC3) complexes participate in the nucleation of the pre-autophagosomal membrane.

**Figure 5 ijms-21-08196-f005:**
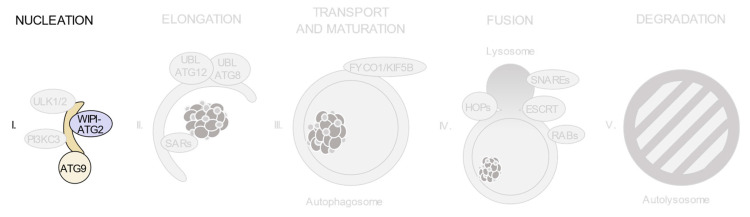
The phosphatidylinositol 3-phosphate (PI(3)P)-binding proteins (WIPI or ATG2 proteins) and ATG9-containing vesicles participate in the nucleation of the pre-autophagosomal membrane.

**Figure 6 ijms-21-08196-f006:**
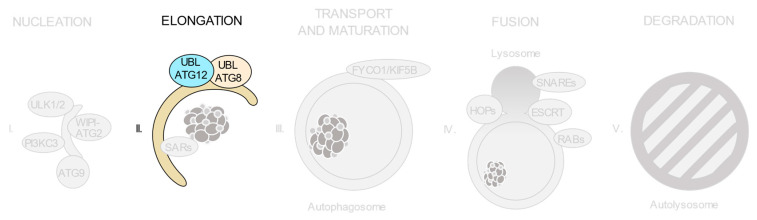
The ubiquitin-like (UBL) conjugation systems of ATG12 and ATG8 participate in the elongation of the pre-autophagosomal membrane.

**Figure 7 ijms-21-08196-f007:**
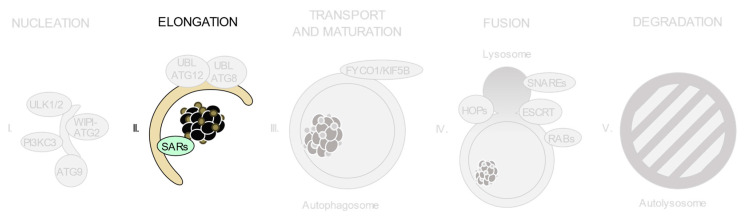
The autophagy receptors (SARs) participate in selective cytoplasmic cargo recognition (i.e., protein aggregates as depicted in the figure) during pre-autophagosomal membrane elongation.

**Figure 8 ijms-21-08196-f008:**
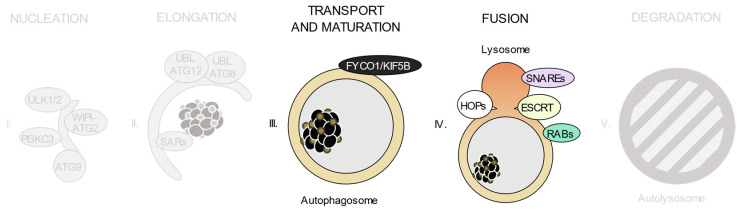
Different cellular machineries (including the HOPs and ESCRT complexes) and effectors (like motor proteins, as well as members of the SNARE or Rab family proteins) are involved in autophagosome transport and maturation, as well as in their fusion with lysosomes.

**Figure 9 ijms-21-08196-f009:**
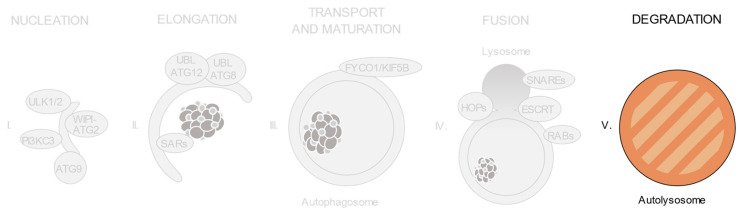
Lysosomal components are direct effectors of autophagosome cargo degradation.

**Table 1 ijms-21-08196-t001:** Clinically relevant SNPs in ULK1/2 complex.

Gene	Disease	dbSNP rsID
*ATG13*	Selective immunoglobulin A deficiency	rs4565870
*ATG13*	Breast cancer	rs10838611
*FIP200*	Hypertension	rs1129660
*ULK1*	Crohn’s disease	rs12303764; rs10902469; rs7488085
*ULK1*	Tuberculosis	rs12297124; rs7138581; rs9481
*ULK1*	Ankylosing spondylitis	rs9652059
*ULK2*	Asparaginase-associated pancreatitis	rs281366

**Table 2 ijms-21-08196-t002:** Clinically relevant SNPs in PI3KC3 complexes.

Gene	Disease	dbSNP rsID
*AMBRA1*	Schizophrenia	rs11819869; rs12574668; rs61882743; rs7112229; rs7130141
*AMBRA1*	Autism	rs3802890
*AMBRA1*	Selective immunoglobulin A deficiency	rs4565870
*ATG14L*	Testicular germ cell tumor	rs1009647
*BECN1*	Machado–Joseph disease	rs60221525
*BECN1*	Diabetes	rs10512488
*UVRAG*	Multiple sclerosis treatment	rs80191572
*UVRAG*	Rheumatoid arthritis	rs7111334
*UVRAG*	Non-segmental vitiligo	rs1458836; rs7933235
*VPS34*	Pancreatic cancer	rs76692125
*VPS34*	Esophageal squamous cell carcinoma	rs52911
*VPS34*	Gastric cancer	rs2162440
*VPS34*	Schizophrenia	rs3813065
*VPS34*	Systemic lupus erythematosus	rs3813065

**Table 3 ijms-21-08196-t003:** Clinically relevant SNPs in PI(3)P-binding proteins and ATG9 orthologues.

Gene	Disease	dbSNP rsID
*ATG2A*	Granuloma formation in Crohn’s disease	rs17146441
*ATG2A*	Hyperuricemia	rs188780113
*ATG2B*	Non-muscle invasive bladder cancer	rs3759601
*ATG2B*	Head and neck squamous cell carcinoma	rs3759601
*ATG9B*	Coronary artery disease	rs2373929; rs7830
*ALFY*	Microcephaly	rs1553924800
*ALFY*	Malignant neoplasm of oropharynx	rs6847067
*DFCP1*	Tuberculosis	rs2333021
*WIPI2*	Osteoporosis	rs4720530
*WIPI3*	Neurodevelopmental disorder	rs786205510; rs1555647262
*WIPI4*	Rett syndrome	rs886041382; rs886041693
*WIPI4*	Neurodegeneration with brain iron accumulation	rs886041382
*WIPI4*	Early-onset epileptic encephalopathy	rs1064793294
*WIPI4*	β-propeller protein-associated neurodegeneration (BPAN)	rs387907330

**Table 4 ijms-21-08196-t004:** Clinically relevant SNPs in the ubiquitin-like conjugation systems ATG12 and ATG8.

Gene	Disease	dbSNP rsID
*ATG10*	Breast cancer	rs10514231; rs1864182; rs7707921
*ATG10*	Paget’s disease of the bone	rs1864183
*ATG10*	Vogt–Koyanagi–Harada syndrome	rs4703863
*ATG10*	Lung cancer	rs10514231; rs1864182; rs1864183; rs10036653
*ATG10*	Melanoma	rs1864182
*ATG10*	Brain metastasis	rs10036653
*ATG10*	Head and neck squamous cell carcinoma	rs10514231; rs1864183; rs4703533
*ATG10*	Pneumoconiosis	rs1864182
*ATG10*	Hepatocellular carcinoma	rs10514231; rs1864183
*ATG12*	Brain metastasis	rs26532
*ATG12*	Pneumoconiosis	rs26538
*ATG12*	Lung cancer	rs26538
*ATG12*	Head and neck squamous cell carcinoma	rs26537
*ATG12*	Hepatocellular carcinoma	rs26537
*ATG16L1*	Crohn’s disease	rs2241880
*ATG16L1*	Palmoplantar pustulosis	rs2241879; rs2241880; rs7587633
*ATG16L1*	Psoriasis vulgaris	rs10210302; rs12994971; rs13005285; rs2241879; rs2241880
*ATG16L1*	Cell-derived thyroid carcinoma	rs2241880
*ATG16L1*	Colorectal cancer	rs2241880
*ATG16L1*	Paget’s disease of the bone	rs2241880
*ATG16L1*	Prostate cancer	rs78835907
*ATG16L1*	Gastric cancer	rs2241880
*ATG16L1*	Melanoma	rs2241880
*ATG16L1*	Brain metastasis	rs2241880
*ATG16L1*	Head and neck squamous cell carcinoma	rs2241880; rs4663402
*ATG16L1*	Lung cancer	rs2241880
*ATG16L1*	Hepatocellular carcinoma	rs4663402
*ATG16L1*	*Helicobacter pylori* infection	rs2241880
*ATG4A*	Cervical Cancer	rs5973822; rs4036579; rs807181; rs807182; rs807183
*ATG4A*	Lung cancer	rs807185
*ATG4A*	Granuloma formation in Crohn’s disease	rs5973822
*ATG4A*	Clear cell renal cell carcinoma	rs7880351
*ATG4B*	Obesity	rs7601000
*ATG4B*	Atherosclerosis	rs139302128
*ATG4C*	Clear cell renal cell carcinoma	rs6670694; rs6683832
*ATG4C*	Kashin–Beck disease	rs11208030; rs4409690; rs12097658; rs6587988
*ATG4D*	Granuloma formation in Crohn’s disease	rs7248036; rs2304165
*ATG5*	Systemic lupus erythematosus	rs6937876; rs3827644; rs573775; rs548234
*ATG5*	Asthma	rs12212740; rs11751513; rs12201458; rs2299863; rs510432
*ATG5*	Parkinson’s disease	rs510432
*ATG5*	Systemic sclerosis	rs3827644; rs9373839
*ATG5*	Non-medullary thyroid cancer	rs2245214
*ATG5*	Neuromyelitis optica	rs548234; rs6937876
*ATG5*	Paget’s disease of the bone	rs2245214
*ATG5*	Behçet’s disease	rs573775
*ATG5*	Spinocerebellar ataxia	rs1131692265
*ATG5*	Crohn’s disease	rs510432; rs9373839
*ATG5*	Multiple myeloma	rs9372120
*ATG5*	Melanoma	rs2245214; rs510432
*ATG5*	Sepsis	rs506027; rs510432
*ATG5*	Pneumoconiosis	rs510432
*ATG5*	Esophageal squamous cell carcinoma	rs1322178; rs3804329; rs671116
*ATG5*	Lung cancer	rs510432; rs688810; rs2245214
*ATG5*	Cerebral palsy	rs6568431
*ATG5*	Breast cancer	rs473543
*ATG5*	Clear cell renal cell carcinoma	rs490010
*ATG5*	Chronic Q fever	rs2245214
*ATG5*	Aplastic anaemia	rs473543; rs510432; rs573775; rs803360
*ATG5*	HBV infection	rs510432; rs6568431; rs548234
*ATG5*	Hepatocellular carcinoma	rs17067724
*ATG7*	Systemic lupus erythematosus	rs11706903; rs2736340
*ATG7*	Breast cancer	rs8154
*ATG7*	Ischemic stroke	rs2594966; rs2594973; rs4684776
*ATG7*	Lung cancer	rs8154
*ATG7*	Clear cell renal cell carcinoma	rs2606736; rs6442260
*ATG7*	Cerebral palsy	rs1470612; rs2594972
*ATG7*	Huntington’s disease	rs36117895
*IRGM*	Crohn’s disease	rs10065172; rs1000113; rs10065172; rs11747270; rs11749391; rs180802994; rs4958843; rs4958847; rs72553867; rs7714584, rs9637876
*IRGM*	Systemic lupus erythematosus	rs10065172; rs13361189
*IRGM*	Ulcerative colitis	rs1000113; rs11747270; rs11749391; rs180802994; rs4958847
*IRGM*	Tuberculosis	rs10051924; rs12654043; rs4958843; rs72553867
*IRGM*	Celiac disease	rs10065172
*IRGM*	Inflammatory bowel diseases	rs10065172; rs4958847
*IRGM*	Ankylosing spondylitis	rs10065172; rs11749391
*IRGM*	Arthritis	rs11747270; rs4958847
*IRGM*	Chronic periodontitis	rs11747270
*IRGM*	Asthma	rs11747270
*IRGM*	Multiple sclerosis	rs11747270
*IRGM*	Cholangitis	rs11749391
*IRGM*	Psoriasis vulgaris	rs11749391
*IRGM*	Pathologic fistula	rs4958847
*IRGM*	Malignant neoplasm of stomach	rs4958847
*IRGM*	Non-alcoholic fatty liver disease	rs4958847
*MAP1LC3A*	Chronic Q fever	rs1040747
*MAP1LC3A*	Coronary artery disease	rs2424994
*MAP1LC3B*	Myopia	rs1054521
*MAP1LC3B*	Systemic lupus erythematosus	rs933717

**Table 5 ijms-21-08196-t005:** Clinically relevant SNPs in selective autophagy receptors.

Gene	Disease	dbSNP rsID
*ATGL*	Neutral lipid storage disease with myopathy	rs121918259
*ATL3*	Hereditary sensory autonomic neuropathy	rs587777108
*BNIP3*	Major depressive disorder	rs9419139
*BNIP3L*	Schizophrenia	rs1042992; rs73219805; rs73219806
*BNIP3L*	Cognitive decline	rs77609452
*HSL*	Lipodystrophy	rs766817317; rs587777699
*HTT*	Huntington’s disease	rs1210554604; rs10015979; rs110501; rs11731237; rs2071655; rs2269499; rs2285086; rs2298969; rs2471347; rs362272; rs363066; rs363092; rs363096; rs3856973; rs6855981; rs82333; rs916171; rs118005095; rs13102260
*NBR1*	Brooke–Spiegler syndrome	rs202122812
*NCOA4*	Prostate cancer	rs10740051; rs10761581
*NCOA4*	Papillary thyroid carcinoma	rs782237788
*NDP52*	Crohn’s disease	rs2303015
*NIPSNAP1*	Breast cancer	rs183421746
*NUFIP1*	Asthma	rs114280567
*OPTN*	Amyotrophic lateral sclerosis	rs267606928; rs267606929
*OPTN*	Primary open-angle glaucoma	rs28939688
*OPTN*	Paget’s disease of the bone	rs1561570; rs2234968
*PHB2*	Bisphosphonate-associated osteonecrosis of the jaw	rs11064477
*FAM134B*	Hereditary sensory autonomic neuropathy	rs137852737; rs137852738; rs137852739; rs886037748
*RTN3*	Malaria	rs542998
*SQSTM1*	Frontotemporal dementia	rs776749939; rs772889843; rs1355424687
*SQSTM1*	Paget’s disease of the bone	rs796051869; rs104893941
*SQSTM1*	Amyotrophic lateral sclerosis	rs796052214; rs796051870; rs796051870
*SQSTM1*	Neurodegeneration	rs886039780
*SQSTM1*	Parkinson’s disease	rs200396166
*SQSTM1*	Atypical apraxia of speech	rs796052214
*SQSTM1*	Sporadic inclusion body myositis	rs11548633
*STBD1*	Parkinson’s disease	rs6812193
*TAX1BP1*	Head and neck carcinoma	rs11540483
*TAX1BP1*	Hypospadias	rs10214930
*TOLLIP*	Leishmaniasis	rs3750920; rs5743899
*TOLLIP*	Leprosy	rs3793964; rs3750920
*TOLLIP*	Malaria	rs3750920
*TOLLIP*	Tuberculosis	rs3750920; rs5743867
*TOLLIP*	Sepsis	rs5743867
*TOLLIP*	Idiopathic pulmonary fibrosis	rs5743890; rs111521887; rs3750920
*TOLLIP*	Fibrotic idiopathic interstitial pneumonia	rs3168046; rs3750920; rs3793964; rs3829223; rs5744034

**Table 6 ijms-21-08196-t006:** Clinically relevant SNPs in cellular machineries involved in autophagosome-lysosome fusion.

Gene	Disease	dbSNP rsID
*BIRC6*	Glaucoma	rs2754511
*C9orf72*	Amyotrophic lateral sclerosis	rs3849943; rs774359; rs3849942
*C9orf72*	Familial frontotemporal dementia with amyotrophic lateral sclerosis	rs71492753
*CHMP2B*	Neuroblastoma	rs63750355; rs63750653
*CHMP2B*	Frontotemporal dementia	rs78268395
*CHMP2B*	Amyotrophic lateral sclerosis	rs281864934
*CHMP4B*	Bilateral cataracts	rs118203966
*CHMP4B*	Diabetes mellitus, non-insulin-dependent	rs7274168
*CHMP4B*	Dysphagia	rs2747539
*EPG5*	Alzheimer’s disease	rs9963463; rs11082498
*EPG5*	Depressive disorders	rs58682566
*EPG5*	Vici syndrome	rs1470797555; rs1555673917; rs1568107449; rs1568112516; rs1568112543; rs1568118775; rs1568133724; rs1568133760; rs201757275; rs587776940; rs587776941; rs587776942; rs762639913; rs767638289; rs780889226; rs863225064; rs866435487; rs961245497; rs863225064
*EPG5*	Cataract	rs201757275
*FYCO1*	Cataract	rs387906963; rs387906964; rs387906965
*HGS*	Age-related macular degeneration	rs8070488
*KIF5B*	Bipolar disorder	rs1775715
*PLEKHM1*	Osteopetrosis	rs786205055
*PLEKHM1*	Parkinson’s disease	rs11012
*PLEKHM1*	Alopecia	rs144733372
*PLEKHM1*	Unipolar depression	rs144733372
*PLEKHM1*	Major depressive disorder	rs144733372
*PLEKHM1*	Ovarian cancer	rs1879586; rs2077606; rs17631303

*RAB33B*	Smith–McCort dysplasia	rs1085307129; rs886044716; rs1085307131; rs1085307128; rs587776958
*RAB7*	Charcot–Marie–Tooth disease type 2B	rs121909080; rs121909078; rs121909079; rs121909081
*SNAP29*	Cednik syndrome	rs387907363; rs869312906
*SPG11*	Spastic paraplegia	rs1085307097; rs118203963; rs140385286; rs1555447432; rs141848292; rs312262720; rs312262721; rs312262722; rs312262737; rs312262749; rs312262752; rs312262764; rs312262779; rs371334506; rs747220413; rs764647588; rs765477482; rs767798272
*STX17*	Alopecia	rs10760706
*STX7*	Neuronal heterotopia	rs864309676
*VAMP8*	Cerebrovascular accident	rs1010
*VAMP8*	Tuberculosis	rs1010
*VAMP8*	Coronary artery disease	rs1010
*VAMP8*	Prostate cancer	rs10187424; rs3731827
*VPS33A*	MPS-like disorder	rs767748011
*VPS41*	Major depressive disorder	rs10274968
*YKT6*	Diabetes	rs2908282
*YKT6*	Birth weight and subsequent risk factors	rs138715366
*ZFYVE26*	Spastic paraplegia	rs1049504575; rs1057518016; rs1214483973; rs1555394376; rs200832994; rs558285072; rs767164213; rs768176054; rs769329153; rs774809466; rs941230062; rs981804211
*ZFYVE26*	Amyotrophic lateral sclerosis	rs12891047
*ZFYVE26*	Breast cancer	rs200595749
*ZFYVE26*	Movement disorders	rs752283089; rs869312914

**Table 7 ijms-21-08196-t007:** Clinically relevant SNPs in lysosomal components.

Gene	Disease	dbSNP rsID
*ARSB*	Mucopolysaccharidosis type VI (Maroteaux–Lamy syndrome)	rs118203938; rs118203939; rs118203940; rs431905493; rs431905495; rs431905496; rs118203942; rs118203944; rs118203943
*CLN3*	Neuronal ceroid lipofuscinosis type 3	rs121434286; rs267606737; rs386833720; rs786201028; rs121434286
*CLN6*	Neuronal ceroid lipofuscinosis type 6	rs104894483; rs104894486; rs121908079; rs121908080; rs397515352; rs774543080; rs786205065; rs786205066; rs786205067; rs104894484
*CLN6*	Adult neuronal ceroid lipofuscinosis	rs154774633; rs154774634; rs154774635; rs154774636
*CLN8*	Neuronal ceroid lipofuscinosis type 8	rs104894060; rs137852883; rs28940569
*CLN8*	Northern epilepsy syndrome	rs104894064
*CTNS*	Cystinosis	rs375952052
*CTSA*	Galactosialidosis	rs137854540; rs137854544; rs137854546; rs137854547; rs137854548; rs137854549; rs786200859; rs875989777; rs137854544; rs137854543
*CTSD*	Neuronal ceroid lipofuscinosis type 10	rs786205105; rs797045137; rs797045138; rs121912789; rs121912790
*CTSF*	Neuronal ceroid lipofuscinosis type 13	rs753084727; rs797045136; rs143889283; rs397514731
*DNAJC5*	Neuronal ceroid lipofuscinosis, Parry type,	rs587776892; rs387907043
*FUCA1*	Fucosidosis	rs118204450; rs80358195; rs80358196; rs80358197; rs80358198
*GAA*	Glycogen storage disease type II (Pompe disease)	rs1057516581; rs12450199; rs140826989; rs121907940; rs121907941; rs1393386120; rs1414146587; rs121907942; rs1344266804; rs121907943; rs121907944; rs1221948995; rs1245412108; rs121907938; rs121907945; rs121907936; rs1800309; rs121907937; rs1800307; rs147804176; rs1555600061; rs1555601773; rs1800312; rs200856561; rs1555601773; rs1800312; rs200856561; rs369531647; rs1057516277; rs886043343; rs892129065; rs28940868;rs1057516215; rs1055945806;
*GAA*	Friedreich ataxia	rs1245992455
*GAA*	Hypochondrogenesis	rs1289257741
*GALNS*	Mucopolysaccharidosis IVA (Morchio A syndrome)	rs1028668536; rs118204438; rs118204449; rs786205899; rs118204435; rs118204441; rs118204442; rs118204446; rs118204447; rs118204448; rs267606838
*GBA*	Parkinson’s disease	rs75548401
*GBA*	Gaucher disease	rs421016
*GLA*	Fabry disease	rs104894828; rs104894834; rs104894845; rs28935197; rs869312142
*GLB1*	GM1 gangliosidosis	rs192732174; rs376663785; rs587776524; rs794727165; rs794729217; rs781658798; rs778423653; rs778700089; rs879050821; rs72555361; rs72555364; rs72555368; rs72555370; rs72555390; rs72555393; rs794729217; rs72555392; rs72555362; rs1214295886; rs1553606128; rs1553610382; rs1553610553; rs1553612189; rs1559401428; rs192732174; rs189115557;
*GLB1*	Mucopolysaccharidosis IVB (Morchio B syndrome)	rs72555363; rs1553606128; rs1553610382; rs1553610553; rs1553612220; rs189115557; rs192732174; rs794729217; rs794727165; rs778700089; rs778423653
*GLB1*	Neuraminidase 1 deficiency	rs1356418704
*GLB1*	Respiratory tract diseases	rs9828592
*GLB1*	Asthma	rs79337446
*GNS*	Mucopolysaccharidosis type IIID (Sanfilippo syndrome)	rs119461974; rs119461975; rs483352898; rs483352899; rs483352900
*GRN*	Presenile dementia	rs373885474
*GRN*	Frontotemporal lobar degeneration	rs606231220; rs63749801; rs63750077; rs63751006; rs63750331; rs63751294; rs63751243
*GUSB*	Mucopolysaccharidosis type VII (Sly syndrome)	rs121918179; rs121918181; rs121918185; rs377519272; rs786200863; rs121918180; rs121918173; rs121918174; rs121918175; rs121918176; rs121918177; rs121918178; rs121918182; rs121918183; rs121918184; rs121918172
*HGSNAT*	Mucopolysaccharidosis type IIIC (Sanfilippo syndrome type C)	rs121908282; rs121908283; rs121908284; rs121908285; rs121908286; rs193066451; rs483352896; rs753355844; rs754875934; rs764206492; rs797045120
*HYAL1*	Mucopolysaccharidosis IX	rs104893743
*IDS*	Mucopolysaccharidosis type II (Hunter syndrome)	rs113993946; rs113993947; rs199422230; rs483352904; rs483352905; rs797044671; rs869025304; rs869025305; rs869025306; rs869025307; rs869025308; rs104894856; rs104894861; rs199422228; rs199422229; rs199422231
*IDUA*	Mucopolysaccharidosis type I (Hurler and Scheie syndrome)	rs121965025; rs121965033; rs199801029; rs387906504; rs398123258; rs762411583; rs786200915; rs869025584; rs121965021; rs121965026; rs121965027; rs121965031; rs121965023; rs121965019; rs121965021; rs121965030; rs764196171; rs121965019; rs121965033; rs121965024;
*LAMP2*	Danon disease	rs104894857; rs104894858; rs1060502302; rs137852527; rs727503118; rs727503119; rs727503120; rs727504742
*MAN2B1*	Alpha-mannosidosis	rs121434331; rs121434332; rs775200333; rs80338677; rs80338678; rs80338679; rs80338680; rs80338681
*MANBA*	Beta-mannosidosis	rs121434334; rs121434335; rs121434336
*MFSD8*	Neuronal ceroid lipofuscinosis type 7	rs11820397; rs587778809; rs724159971; rs727502801; rs118203975; rs118203976; rs140948465; rs267607235; rs749704755
*MFSD8*	Late-infantile neuronal ceroid lipofuscinosis	rs200319160
*NAGLU*	Mucopolysaccharidosis type IIIB (Sanfilippo syndrome type B)	rs104894591; rs104894592; rs104894597; rs104894598; rs118204025; rs746006696; rs886039894; rs886039895; rs118204024; rs104894590; rs104894593; rs104894594; rs104894595; rs104894597; rs104894598; rs753520553; rs796052122; rs104894601
*NPC1*	Niemann–Pick disease, type C	rs1055204017; rs1057518711; rs1474434210; rs753768576; rs139751448; rs143124972; rs28942104; rs756815030; rs758231839; rs886042270; rs80358257; rs80358254; rs80358259; rs150334966; rs1555634422; rs768999208; rs80358259
*NPC2*	Niemann–Pick disease type C	rs80358262; rs80358263; rs80358266; rs80358268; rs11694; rs80358261; rs80358264; rs104894458
*SGSH*	Mucopolysaccharidosis type IIIA (Sanfilippo syndrome type A)	rs104894635; rs104894637; rs138504221; rs1057521801; rs374621913; rs770947426; rs777956287; rs778700037; rs104894638; rs104894640; rs104894642; rs104894643; rs104894636; rs104894641; rs138504221; rs104894635
*SLC17A5*	Sialic acid storage diseases (SASDs)	rs386833987; rs386833994; rs727504156; rs119491109; rs119491110; rs80338795; rs80338794

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
