# Peer review of "Pathogenic Single Nucleotide Polymorphisms on Autophagy-Related Genes"

_ijms, 2020, doi:10.3390/ijms21218196_

Round 1
Reviewer 1 Report
In this manuscript Tamargo-Gomez et al. nicely describe recent evidence on the potential pathogenic role of already described SNPs occurring in autophagy-associated genes. The review is very well written, interesting, timely, and with relevance for people working on different fields. I only have a few MINOR comments that could improve the final version of the manuscript:
- Figure 2 quality is very poor. I guess it may be something related to the uploading of the manuscript during the submission, but that must certainly be corrected in the final version of the manuscript.
- In the past 8-10 years there is an increasing number of manuscripts describing how LSD-associated mutations affecting the levels of lysosomal lipids can disrupt autophagy by mechanisms such as lysosomal permeabilization (PMID: 24488099), impaired lysosomal proteolysis (PMID: 22872701), impeded autophagosomal closure (PMID: 27070082), or blockade of lysosome to autophagosome fusion (PMID: 20871593, PMID: 17913701). Because of this growing evidence of papers linking lysosomal lipid imbalance and autophagy dysfunction is now well accepted, I do think these papers should be commented on the “Lysosomal components” part.
- Line 411 states “Although it might not be considered a bona fide selective autophagy receptor, HTT can act as an scaffold for selective autophagy and interacts with ULK1, GABARAP and p62/SQSTM1 [193]”. A couple of points in this sentence; “act as an scaffold” should be “act as a scaffold” instead. Additionally, here I miss a couple of important papers, that should be included, on the role that huntingtin could play as an autophagy receptor (PMID: 25686248, PMID: 20383138).
- Line 50, “as” is duplicated.
Apart from that, I think the authors have made a good effort and a good job on putting together a considerable amount of previous research on the topic and therefore I can recommend the publication of the manuscript once these minor suggestions/issues are included.
Author Response
POINT-BY-POINT REPLY TO REFEREE NO. 1
Reviewer #1 commented:
In this manuscript Tamargo-Gomez et al. nicely describe recent evidence on the potential pathogenic role of already described SNPs occurring in autophagy-associated genes. The review is very well written, interesting, timely, and with relevance for people working on different fields. I only have a few MINOR comments that could improve the final version of the manuscript:
- Figure 2 quality is very poor. I guess it may be something related to the uploading of the manuscript during the submission, but that must certainly be corrected in the final version of the manuscript.
We would like to acknowledge the reviewer’s commentary about the quality of the figure. As he/she points out, the low quality is due to the necessary compression for uploading the initial manuscript. We have assured that the final version of that and the other manuscript’s figures has the best possible quality
- In the past 8-10 years there is an increasing number of manuscripts describing how LSD-associated mutations affecting the levels of lysosomal lipids can disrupt autophagy by mechanisms such as lysosomal permeabilization (PMID:24488099), impaired lysosomal proteolysis (PMID: 22872701), impeded autophagosomal closure (PMID: 27070082), or blockade of lysosome to autophagosome fusion (PMID: 20871593, PMID: 17913701). Because of this growing evidence of papers linking lysosomal lipid imbalance and autophagy dysfunction is now well accepted, I do think these papers should be commented on the “Lysosomal components” part.
We would like to acknowledge the reviewer’s suggestions on including additional references covering how LSD-associated mutations affecting the levels of lysosomal lipids can disrupt autophagy. We have included all the suggested references in the text
- Line 411 states “Although it might not be considered a bona fide selective autophagy receptor, HTT can act as an scaffold for selective autophagy and interacts with ULK1, GABARAP and p62/SQSTM1 [193]”. A couple of points in this sentence; “act as an scaffold” should be “act as a scaffold” instead. Additionally, here I miss a couple of important papers, that should be included, on the role that huntingtin could play as an autophagy receptor(PMID: 25686248, PMID: 20383138).
We have corrected the mentioned grammatical error. In addition, we have cited the suggested references describing the role of huntingtin as an autophagy receptor
- Line 50, “as” is duplicated.
We have corrected that typo. We thank the reviewer for this correction
Reviewer 2 Report
General comment
This manuscript summarizes the relationship between SNPs polymorphism and diseases of autophagy-related genes. This manuscript is a review article that extensively covers the literature so far, and fully covers the targets well. On the other hand, the impression was also strong as a comprehensive descriptive review of reports on the possible relationship between SNPs of each autophagy factor and disease. This may be due to the lack of brief conclusions, or descriptions of limitations and problems, at the end of each section (3.1-3.7). I think that as much description as possible is required. If such a statement is possible, it will clearly brush up this review.
Major comments
- Figure 2 is very well organized visually. However, if the additional interpretations appear in the text, it will be very helpful to the reader.
- Reference numbers should be added to the 4th column of each table.
- In addition to the upper part of Fig.2, if possible, could a scheme of autophagy mechanism be illustrated for each section or common to each section? This dramatically advances the reader's understanding.
Minor comments
- Line 199, line 201: PI3KC3 and PI(3)P, the abbreviations for each are already on lines 173 and 174.
- In my impression, how about thinking as the last sentence as follows? Lines 562-564: For this reason, the identification of new clinically significant SNPs is not only important in terms of disease prevention, but also to design new therapeutic strategies aimed at modulating autophagy for clinically-relevant purposes. This review should be of great help in advancing to design new therapeutic strategies.”
Author Response
This manuscript summarizes the relationship between SNPs polymorphism and diseases of autophagy-related genes. This manuscript is a review article that extensively covers the literature so far, and fully covers the targets well. On the other hand, the impression was also strong as a comprehensive descriptive review of reports on the possible relationship between SNPs of each autophagy factor and disease. This may be due to the lack of brief conclusions, or descriptions of limitations and problems, at the end of each section (3.1-3.7). I think that as much description as possible is required. If such a statement is possible, it will clearly brush up this review.
We have included a new paragraph at the end of each section summarizing the clinical significance of the SNPs covered in each of the different sections. We have include the type of pathologies these SNPs are associated with, discussing when there is any trend towards the development of any specific group of pathologies caused by specific SNPs in a given gene or group of autophagy-related molecular components. We acknowledge the reviewer’s suggestion, as we think that including these final paragraphs add more clarity to the text.
Major comments
- Figure 2 is very well organized visually. However, if the additional interpretations appear in the text, it will be very helpful to the reader.
We have included additional schemes for the upper part of the figure in each of the sections in the text (as requested in point#3), linking each of them with the molecular complexes or stage of autophagic degradation that each paragraph deals with. Moreover, we have included a brief closing paragraph in each of the sections in which we add information on the type of diseases caused by the SNPs covered in each of the different sections, thus adding additional interpretation for figure 2 information throughout the text.
- Reference numbers should be added to the 4th column of each table.
Due to the high number of references covered in our manuscript, we think that although it is clearly a good idea to include reference numbers in the 4th column of the table, by doing so, the number of references in the manuscript would be excessively huge, given that the manuscript has 343 references. Instead, we have included the PMIDs for all the references covering the SNPs described in the manuscript in supplementary tables.
- In addition to the upper part of Fig.2, if possible, could a scheme of autophagy mechanism be illustrated for each section or common to each section? This dramatically advances the reader's understanding.
Following the reviewer’s suggestion, we have included additional figures illustrating which part of the autophagy mechanism is referred to in each section. We acknowledge the reviewer’s suggestion, as we think that including these specific figures might help readers to better understand and follow the manuscript
Minor comments
- Line 199, line 201: PI3KC3 and PI(3)P, the abbreviations for each are already on lines 173 and 174.
We have corrected that mistake. We thank the reviewer for this comment
- In my impression, how about thinking as the last sentence as follows? Lines 562-564: For this reason, the identification of new clinically significant SNPs is not only important in terms of disease prevention, but also to design new therapeutic strategies aimed at modulating autophagy for clinically-relevant purposes. This review should be of great help in advancing to design new therapeutic strategies.”
We would like to acknowledge the reviewer’s commentary on that particular sentence and have rewritten it as suggested. We feel that this change improves how our manuscript concludes.